# Changes in transmission of Enterovirus D68 (EV-D68) in England inferred from seroprevalence data

**Margarita Pons-Salort[1]\*, Ben Lambert[2], Everlyn Kamau[3], Richard Pebody[4], Heli Harvala[5], Peter Simmonds[3], Nicholas C Grassly[1]**

[1]MRC Centre for Global Infectious Disease Analysis, School of Public Health, Imperial College London, London, United Kingdom; [2]Department of Computer Science, University of Oxford, Oxford, United Kingdom; [3]Nuffield Department of Medicine, University of Oxford, Oxford, United Kingdom; [4]Immunization Department, Public Health England, London, United Kingdom; [5]Infection and Immunity, University College of London, London, United Kingdom

**Abstract** The factors leading to the global emergence of Enterovirus D68 (EV-D68) in 2014 as a cause of acute flaccid myelitis (AFM) in children are unknown. To investigate potential changes in virus transmissibility or population susceptibility, we measured the seroprevalence of EV-D68-specific neutralising antibodies in serum samples collected in England in 2006, 2011, and 2017. Using catalytic mathematical models, we estimate an approximately 50% increase in the annual probability of infection over the 10-year study period, coinciding with the emergence of clade B around 2009. Despite such increase in transmission, seroprevalence data suggest that the virus was already widely circulating before the AFM outbreaks and the increase of infections by age cannot explain the observed number of AFM cases. Therefore, the acquisition of or an increase in neuropathogenicity would be additionally required to explain the emergence of outbreaks of AFM. Our results provide evidence that changes in enterovirus phenotypes cause major changes in disease epidemiology.

**\*For correspondence:** m.pons-salort@imperial.ac.uk

**Competing interest:** The authors declare that no competing interests exist.

## Editor's evaluation

The authors use data from three cross-sectional age-stratified serosurveys on Enterovirus D68 from England between 2006 and 2017 to examine the transmission dynamics of this pathogen. This study's convincing methodology provides valuable insights into the changing dynamics of enterovirus D68, uncovering potential changes in the transmissibility of the virus. It will be of interest to infectious disease epidemiologists and surveillance professionals.

## Introduction

Interest in understanding the epidemiology and disease impact of Enterovirus D68 (EV-D68) and other enteroviruses has increased in recent years. Contrary to most human enteroviruses, EV-D68 causes severe respiratory disease and is transmitted by the respiratory route, sharing properties with rhinoviruses (*Oberste et al., 2004*). Although this virus was first isolated in 1962, for decades it was only reported from isolated cases or small case clusters of respiratory disease (*Pons-Salort et al., 2015*).

From 2009 to 2010 onwards however, an increasing number of outbreaks of EV-D68-associated severe respiratory illness have been reported worldwide (*Holm-Hansen et al., 2016*; *Tokarz et al., 2012*). In 2014, the US experienced the first big outbreak of respiratory disease linked to EV-D68, with >1100 cases reported by the Centers for Disease Control and Prevention (*Midgley et al., 2015*). In

parallel with this outbreak, an unusual number of acute flaccid myelitis (AFM) cases (a newly recognised condition that includes the sudden onset of flaccid limb weakness [*Centers for Disease Control and Prevention, 2021*]) were also reported, and similar AFM outbreaks subsequently occurred in 2016 and 2018 (*Park et al., 2021*). Retrospectively, it now appears, an unusual spike of 'polio-like' cases reported in 2012 in California (*Ayscue et al., 2014*) was an early occurrence of what was subsequently defined as AFM. In the UK and elsewhere in Europe, AFM cases have also been reported in recent years associated with upsurges in EV-D68 detections (*Knoester et al., 2019*; *The United Kingdom Acute Flaccid Paralysis (AFP) Task Force, 2019*; *Williams et al., 2016*). Evidence that EV-D68 is the main cause of these AFM outbreaks has been growing (*Park et al., 2021*; *Messacar et al., 2018*), although the role of other enterovirus serotypes such as enterovirus A71 (EV-A71) has not been discounted (*McKay et al., 2018*). There is no effective treatment or vaccine for EV-D68 infection yet, and residual paralysis and neurological sequelae after AFM is common and lifelong.

The mechanisms that have led to the emergence of EV-D68 outbreaks since the late 2000s remain unknown. One hypothesis is that transmission has increased as a result of evolutionary selection for increased replication fitness, or through the appearance of immune escape-associated mutations that lead to the evasion of pre-existing population immunity. Another is that the virus has become more pathogenic, and, as a consequence, the number of symptomatic (and therefore, reported) infections has increased independently of its transmissibility (i.e. the virus already circulated in the past but went mostly undetected).

As for other enterovirus serotypes, many EV-D68 infections are asymptomatic or mild and self-limiting. In addition, enterovirus surveillance is passive in most countries, based on laboratory reporting for samples submitted by clinicians for testing. It is consequently difficult to determine the true incidence of infection or whether changes in EV-D68 circulation have occurred. A recent study based on data from the BioFire FilmArray Respiratory Panel (*Meyers et al., 2020*) has shown biennial cycles of EV-D68 circulation in the US at the national level since 2014, coinciding with the years of AFM outbreaks (*Park et al., 2021*). Similarly, in the UK, reported EV-D68 virus detections also show a biennial pattern between 2014 and 2018 (*Figure 1—figure supplement 1*). However, these data are limited before 2014 and respiratory samples or throat swabs are infrequently tested by enterovirus surveillance programmes in the US, the UK or elsewhere. Seroprevalence surveys therefore offer an attractive potential alternative opportunity to investigate patterns of exposure to EV-D68. Detection of EV-D68 antibodies with adequate sensitivity and specificity can indicate prior infection and can be analysed using mathematical models to infer trends in the incidence of infection over time, by age-group and location.

Here, we use data on the prevalence of neutralising antibodies against EV-D68 from opportunistically collected serum samples broadly representative of the general population in England in 2006, 2011, and 2017 to reconstruct long-term changes in EV-D68 transmission. Using a mathematical model-based framework, we estimate changes in the annual force of infection (FOI) and reconstruct the estimated annual number of new infections in each age class.

## Results

Individuals are assigned an antibody titre as the highest antibody dilution (1:4, 1:8, …, 1:2048) preventing virus replication (i.e. showing neutralisation) (*Kamau et al., 2019*). For EV-D68, it is unknown which neutralising antibody titre (or seropositivity cut-off) is indicative of true past infection. A simple method to determine a seropositivity cut-off is based on fitting a mixture model to the individual titre distribution, in order to differentiate between two sub-populations (seronegatives and seropositives) (*Hens et al., 2012*). However, determining such a cut-off was not possible here, as for two of the three serosurveys and for all the data combined, the distributions did not show a bi-modal shape (*Figure 1—figure supplement 2*). We therefore present our modelling analysis for two different cut-offs: a first weak cut-off of 1:16, which has been previously used in the literature to define EV-D68 seropositivity (*Kamau et al., 2019*; *Karelehto et al., 2019*), and a more stringent cut-off of 1:64, which provides seroprevalence curves by age similar to an even more stringent cut-off of 1:128 (*Figure 1—figure supplement 3*).

Seroprevalence frequencies in different age groups from the three serosurveys of samples collected in 2006, 2011, and 2017 are shown in *Figure 1*. At each time point, irrespective of the cut-off antibody titre chosen to define seropositivity, seroprevalence slightly decreases from the 0 years-old (yo) to the

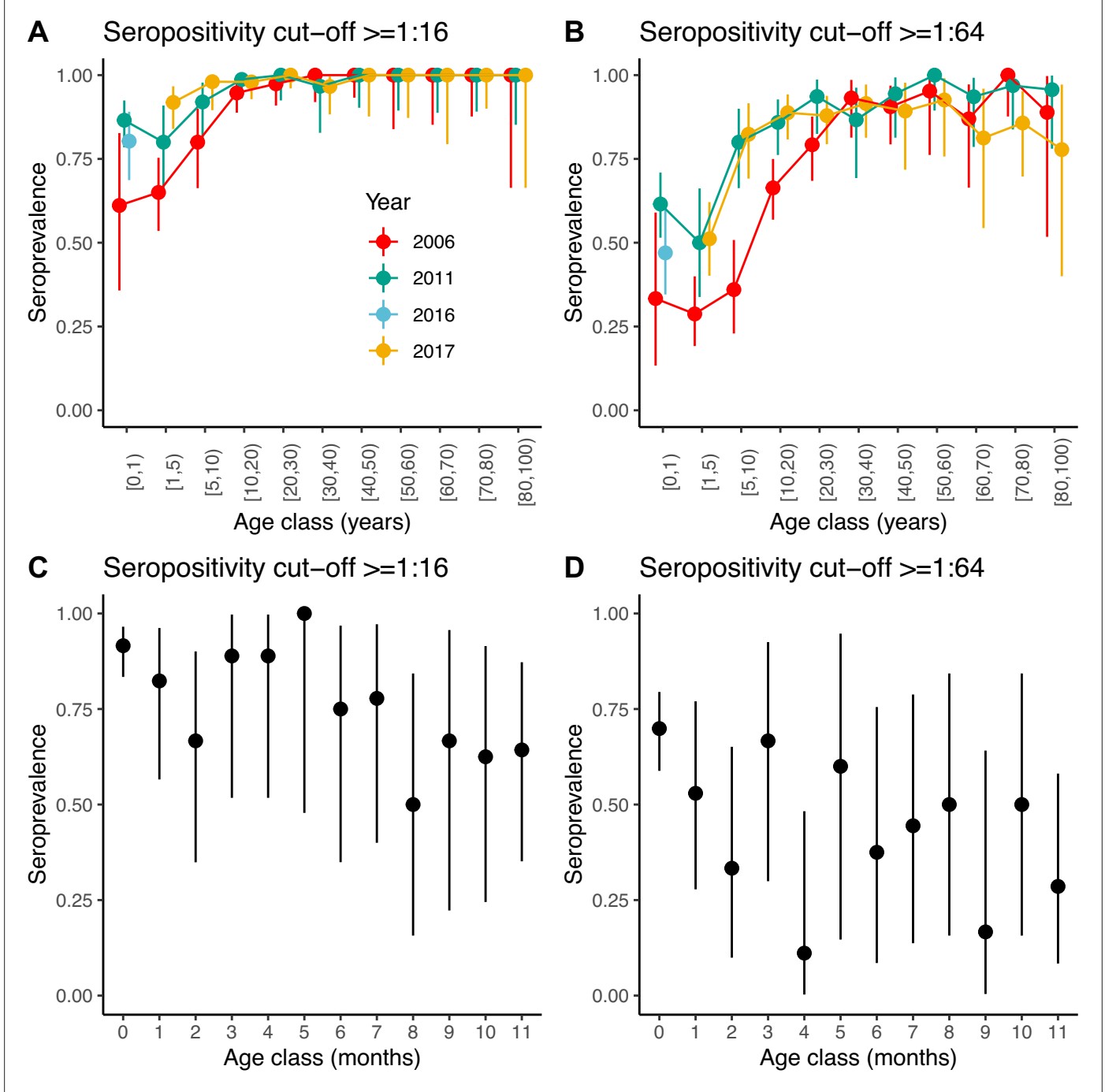

**Figure 1.** Seroprevalence by age. Seroprevalence by age class in years for the three serosurveys and for two different cut-offs of neutralising antibody titre used to define positivity: (**A**) 1:16 and (**B**) 1:64. Seroprevalence by month during the first year of life for children across the three serosurveys combined and two seropositivity cut-offs: (**C**) 1:16 and (**D**) 1:64. Bars are 95% binomial confidence intervals. Note that in (**A**) and (**B**) individuals in the [0–1] age class from the 2017 serosurvey were in fact sampled in 2016 and as such, are shown with a different colour.

The online version of this article includes the following figure supplement(s) for figure 1:

**Figure supplement 1.** Annual number of Enterovirus D68 (EV-D68) detections reported by Public Health England (now UKHSA) between 2004 and 2020.

**Figure supplement 2.** Individual log2 titre distributions.

**Figure supplement 3.** Seroprevalence by age class in years for the three serosurveys and two different cut-offs of neutralising antibody titre used to define positivity: (**A**) 1:32 and (**B**) 1:128.

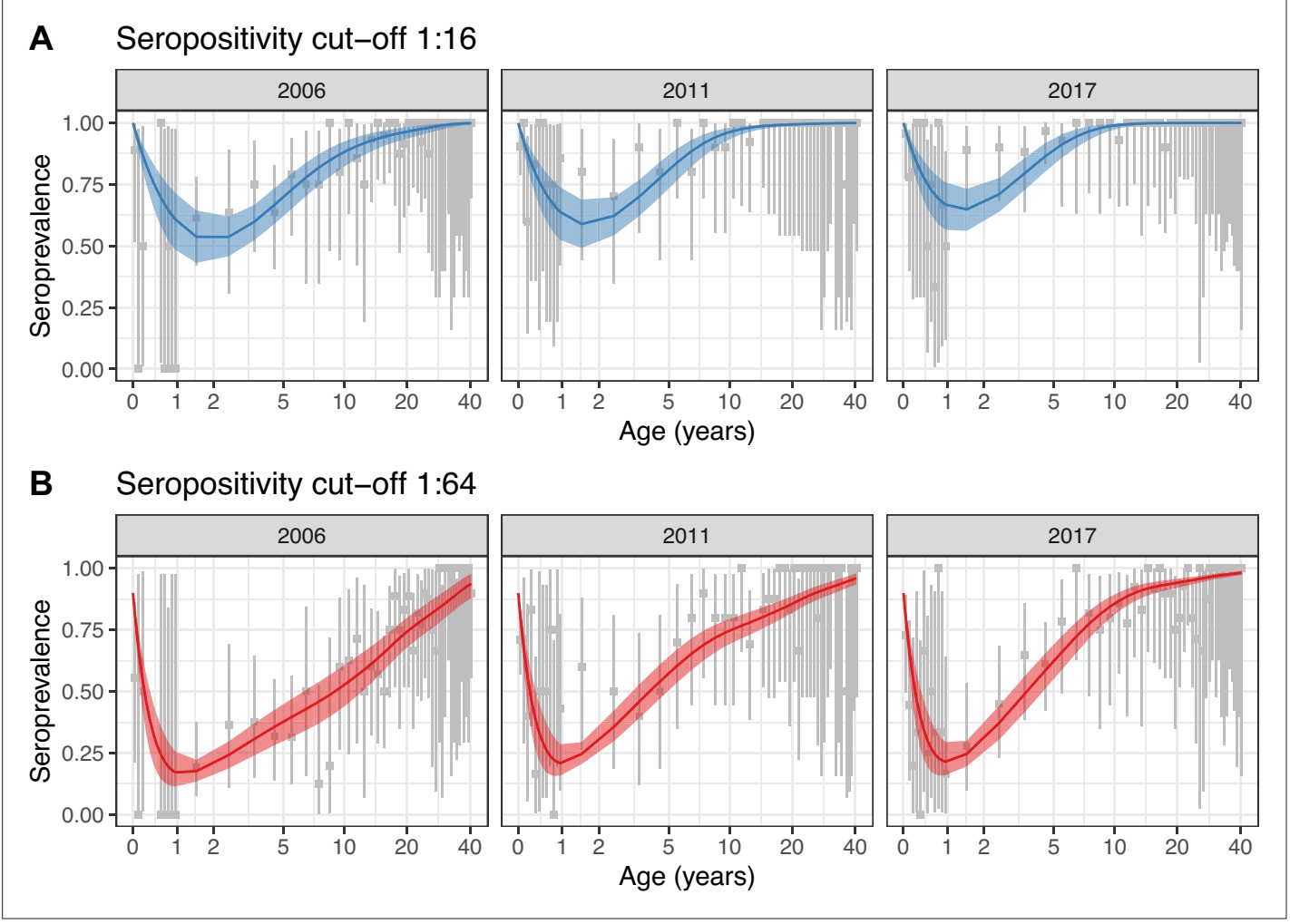

**Figure 2.** Best model fit to data. Observed (gray) and estimated (blue and red) seroprevalence by age for the three cross-sectional serosurveys (2006, 2011, and 2017). Model fit is shown for the random walk model (Model 2) using the two seropositivity cut-offs, (**A**) 1:16 and (**B**) 1:64. Gray intervals indicate the 95% binomial confidence intervals around the observed proportion of seropositivity. The line and ribbons correspond to the median and 95% credible intervals of the posterior estimates of seroprevalence. Note that the age axis is log-transformed to better show the data in the younger age classes. The same plot with the age axis on a natural scale is shown in *Figure 2—figure supplement 2*. Note also that the data for <1 yo shown for 2017 were actually collected in 2016.

The online version of this article includes the following figure supplement(s) for figure 2:

**Figure supplement 1.** Constant model fit to data.

**Figure supplement 2.** Same as *Figure 2*, with *x*-axis on a natural scale.

**Figure supplement 3.** Same as *Figure 2—figure supplement 1*, with *x*-axis on a natural scale.

1–4 yo age classes, and then increases sharply with age until the 20–29 yo, when it reaches a plateau (*Figure 1A, B*). As for many other viruses, higher values in the 0-yo age class are likely the result of the presence of transplacentally acquired maternal antibodies that subsequently decline. For a seropositivity cut-off of 1:16, the proportion seropositive at ages 1–4 yo ranged between 0.65 (95% confidence interval [CI] 0.54–0.75) in 2006 and 0.92 (95% CI 0.84–0.97) in 2017. For a more stringent cut-off of 1:64, the proportion seropositive in this age group decreased to 0.29 (95% CI 0.19–0.40) in 2006 and 0.51 (95% CI 0.40–0.62) in 2017. Age-stratified seroprevalence was generally lower in 2006 compared to 2011 and 2017, which suggests individuals acquired their first infection at a lower age through the study period, leading to a decrease in the mean age of exposure. This could potentially be consistent with increased transmission (e.g. through increased viral fitness or the accumulation of susceptible) or other mechanisms (such as a change in the virus to have a higher tendency to infect children).

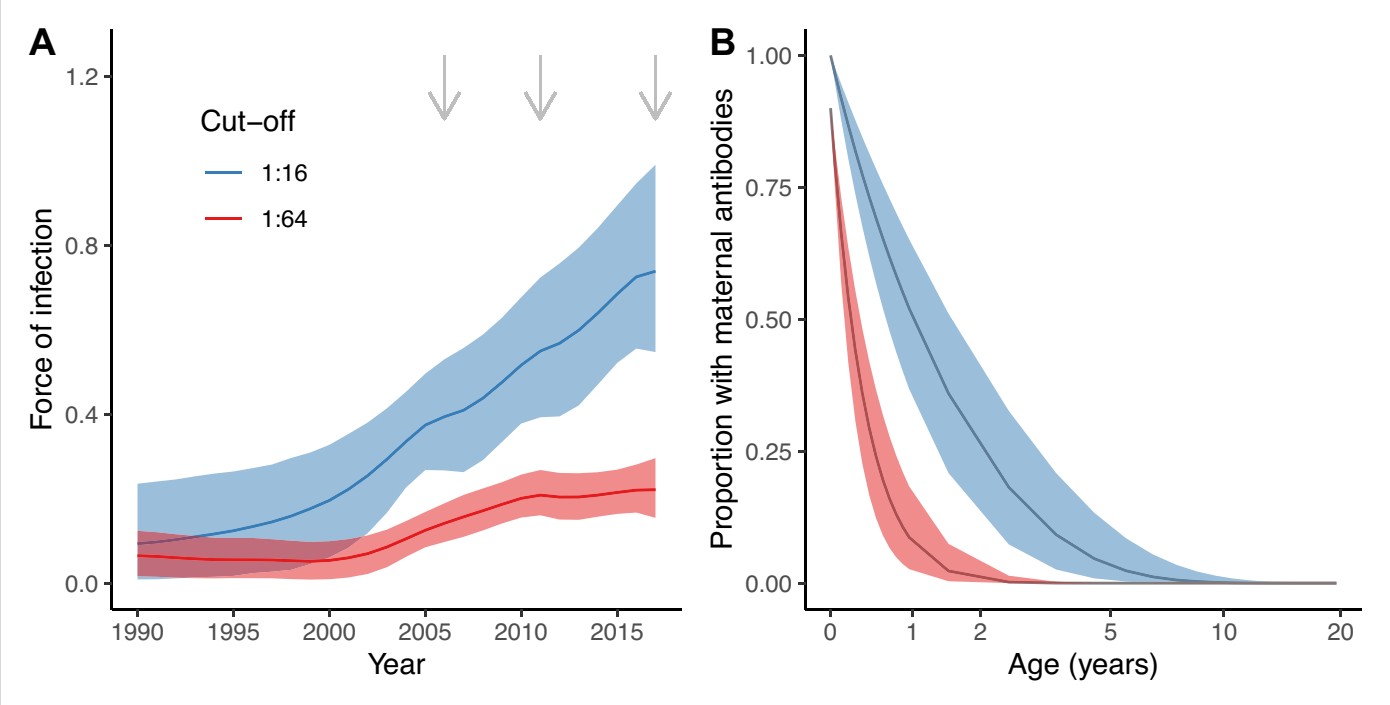

**Figure 3.** Estimates from the best model (Model 2). Estimated force of infection (FOI) over time (**A**) and estimated proportion of individuals with detectable maternal antibodies through age (**B**) for the two seropositivity cut-offs considered. Median and 95% credible intervals are shown for both. In (**A**), gray arrows indicate the years for which there is cross-sectional seroprevalence data.

We also explored seroprevalence during the first year of life using data for all children <1 yo across the three serosurveys combined (*Figure 1C, D*). For both cut-offs, seroprevalence in these children starts very high (92% and 70% for the 1:16 and 1:64 cut-offs, respectively, for children <1 month of age) and declines with age, with a quicker decline during the first 3 months of life then followed by a plateau or slower decline. However, the small number of children per age class resulted in large 95% CIs around the mean. Seroprevalence reaches a minimum at 8 and 4 months of age for the 1:16 and 1:64 cut-offs, respectively. However, the minimum seroprevalence remains high at 50% for the 1:16 cut-off, compared to only 11% for the 1:64.

To address the question of whether transmission had increased before the first reported big outbreaks of EV-D68 in 2014, we compared the performance of two catalytic models that differed on the assumptions of how the FOI changed over time (Materials and methods). Both models assumed that a proportion of individuals are born with maternal antibodies that will subsequently decline at a constant rate, and assumed the risk of infection (becoming seropositive) was independent of age. Model 1 assumed that the FOI was constant over time, and Model 2 allowed it to vary following a random walk. The proportion of individuals born with maternal antibodies was fixed to 1.0 and 0.9 for the seropositivity cut-offs of 1:16 and 1:64, respectively, based on the proportion of adults 25–40 years old that were seropositive in the three serosurveys (*Supplementary file 1a*).

For both seropositivity cut-offs, Model 2 was the best model according to the leave-one-out (LOO) information criterion (see Materials and methods), which accounts for over-parameterisation (*Supplementary file 1b*). Model 2 also provided a better fit to the data (*Figure 2*) than Model 1, which did not capture the observed general increase in seroprevalence between 2006 and 2011 (*Figure 2—figure supplement 1*). All parameter estimates are presented in *Supplementary file 1c*.

The best model (Model 2) estimated an increase in transmission over time during the study period (2006–2017) for both seropositivity cut-offs, as shown by the estimated FOI in *Figure 3A*. For the cut-off of 1:16, the FOI continued to increase until the end of the study period, in 2017. However, for the more stringent cut-off of 1:64, the FOI plateaued from around 2011. These differences reflect the differences in seroprevalence observed in the young age classes (1–20 yo) between 2011 and 2017 for the two cut-offs (*Figure 1*).

There were also important discrepancies in the estimates of the proportion of individuals with maternally acquired antibodies during the first years of life depending on the seropositivity cut-off (*Figure 3B*). We estimated a quicker decline of maternal antibodies for the more stringent cut-off (1:64), with an estimated median duration of seropositivity due to the presence of maternal antibodies of 4.9 (95% CrI 3.3–7.2) months compared to 17.6 (95% CrI 11.5–26.8) months for the 1:16 cut-off (*Figure 3B*). Assuming 90% of individuals are born with maternally acquired antibodies for the 1:64 cut-off, this results in a proportion of only 2.3% (95% CrI 0.4–7.4) of individuals being seropositive due to the presence of maternal antibodies among the 1 yo, compared to 36% (95% CrI 21–51%) for the 1:16 cut-off and 100% of individuals assumed to be born seropositive.

We next used the parameter estimates from the best model and data on the age structure of the population to reconstruct the overall annual EV-D68 seroprevalence in the population (*Figure 4*) for the period between the first and last cross-sectional serosurveys, 2006–2017. For the two sero-positivity cut-offs, the overall seroprevalence was already very high in 2006 (91% [95% CrI 89–93] for 1:16 and 70% [66–74] for 1:64, respectively), and continued to increase progressively until 2017, reaching 97% (96–98%) and 87% (85–89%), respectively, for the 1:16 and 1:64 cut-offs. The detail of the progressive increase in seroprevalence by age year over year as modelled by the random walk is shown in *Figure 5A, B*.

The amount of transmission or extent of virus circulation is better quantified by the number of infections than the FOI, which is sensitive to changes in the age structure of the population (e.g. driven by changes in birth rates) (*Tan et al., 2019*). Using the best model, we reconstructed the annual number of infections in each age class over time (*Figure 5C, D*). For both seropositivity cut-offs, the 2-yo age class has the highest annual incidence of infection. The increase in the FOI between 2006 and 2017 results in an increase in the number of infections in the youngest age classes over time, and a decrease in the oldest, with and inflection point around the age of 4–6 yo. This, in turn, results in a decrease in the mean age at infection from 8.5 yo (7.0–10.4) in 2007 to 4.2 yo (3.5–5.1) in 2017 for the 1:16 cut-off, and from 14.3 yo (12.9–15.9) to 9.8 yo (8.9–10.8) for the 1:64.

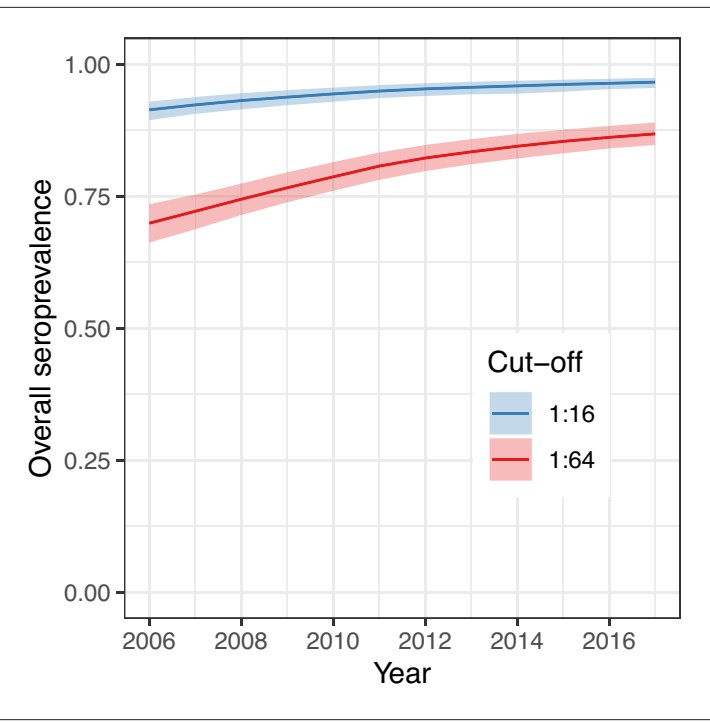

**Figure 4.** Overall (age-weighted) seroprevalence per year for the best model using the two different seropositivity cut-offs. Median and 95% credible intervals are shown.

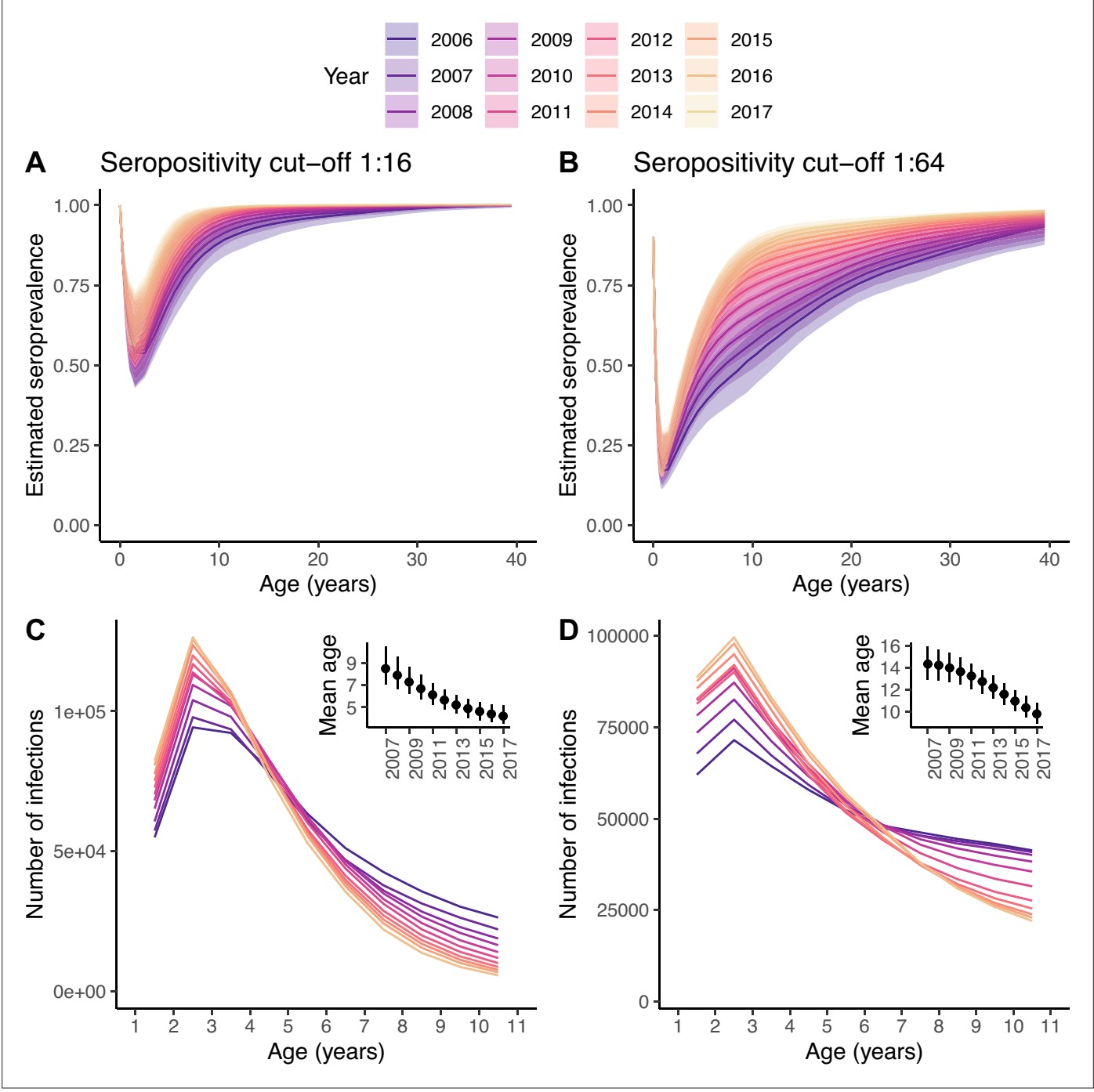

**Figure 5.** Changes in the number of infections. Seroprevalence (**A, B**) and reconstructed number of infections (**C, D**) for each age class and year under the best model and for the two seropositivity cut-offs, 1:16 (**A, C**) and 1:64 (**B, D**). The insets in (**C, D**) show the mean age at infection for each year. In (**A, B**), the median and 95% credible intervals are shown, whereas in (**C, D**) only the median is shown, for clarity of the plots.

## Discussion

The model-based analysis of individual serological data for EV-D68 from three time points (2006, 2011, and 2017) in England presented here suggests an increase in transmission of EV-D68 that occurred or started before 2011. This coincides with the increased number of outbreaks of EV-D68-associated severe respiratory diseases reported worldwide since the late 2000s (*Pons-Salort et al., 2015*; *Tokarz et al., 2012*).

Although our results of an increase in FOI over the study period are robust to the choice of the seropositivity cut-off, we find striking differences in terms of the magnitude of the FOI and the decline of maternal antibodies depending on the seropositivity cut-off used. The results obtained with the more stringent cut-off of 1:64 seem more realistic, both for the estimated annual probabilities of infection (range 0.32–0.52 for the 1:16, and 0.13–0.20 for the 1:64, for the period 2006–2017) and rate of decline of maternal antibodies (average duration of maternal antibodies around 18 mo for the 1:16, and 5 mo for the 1:64). Furthermore, the analysis with the 1:64 cut-off infers that there has not been significant changes in the FOI since 2011; however, with the 1:16 cut-off, the FOI continues to increase until 2017. These results therefore raise the question of what is a suitable seropositivity cut-off to define previous exposure to EV-D68. That said, most EV-D68 seroepidemiology studies published to date present results for a cut-off of 1:8 (as classically used for polioviruses) or 1:16 (*Karelehto et al., 2019*; *Harrison et al., 2019*; *Sun et al., 2018a*; *Xiang et al., 2017*).

One strength of our analysis is that we account for the decline of maternal antibodies in the catalytic models and are able to estimate the rate of this decline, thanks to the availability of data in the <1 yo. Only one study has reported detailed seroprevalence in the <1 yo (*Sun et al., 2018b*). This study was conducted in China in 2010, and the data reported are for a seropositivity cut-off of 1:8. Although they found 100% seroprevalence in neonates, close to the observations in our study when using the 1:16 cut-off, this declined to only 28% in the 9–11 months old, which is much lower than the 64% in the same age class in our dataset (*Figure 1C*). The discrepancy may be partly due to differences in the time of data collection. Indeed, we show data for the three cross-sectional serosurveys combined (2006, 2011, and 2017), given that the number of individuals in each age class was very small, but data for individual years suggest lower seroprevalence in 2006 compared to 2011 and 2017 in the 9–11 mo (*Figure 2A*).

Our findings point to a clear increase in transmission as measured by the estimated annual probability of infection (approximately, 50% higher in 2017 compared to 2006). However, the reconstructed annual number of new infections in each age class suggests that this increase is mostly driven by an increase in the total number of infections in children aged 1–5 yo. Increased transmission in the youngest age groups may be consistent with observed data showing higher and increasing numbers of respiratory illnesses associated with EV-D68 in these age groups (<5 yo) (*Bubba et al., 2020*). However, these results on their own are unlikely to explain the worldwide emergence of AFM outbreaks reported since 2014. First, the high overall seroprevalence observed in 2006 suggests that EV-D68 was already widely circulating before 2014. Second, AFM cases do not exclusively or predominantly occur in the youngest age groups. In the US, for example, AFM cases reported in 2014 had a median age of 7.1 yo (interquartile range [IQR], 4.8–12.1 yo) (*Sejvar et al., 2016*). In the UK, a study from 2018 reported 40 cases, of which only 22 were 0–5 yo (*The United Kingdom Acute Flaccid Paralysis (AFP) Task Force, 2019*). A study from European countries (2016) reported a median age of cases of 3.8 yo and a range of 1.6–9.0 yo. In Japan, a case series study of an AFM cluster reported an overall median age of cases of 4.4 yo (IQR, 2.6–7.7 yo) (*Chong et al., 2018*). Although susceptibility to EV-D68-related AFM may vary with age (*Hixon et al., 2019*) (making it difficult to make a link between the inferred number of infections by age class and the observed number of cases by age), our model-based results suggest that incidence of infections in some of the age groups affected by AFM has not increased, but rather decreased, despite the general increase in transmission. Therefore, the acquisition of or an increase in neuropathogenicity (independent of the described increase in transmission) seems necessary to explain the emergence of AFM through an as-yet unidentified biological mechanism.

The prediction of an increase in the number of infections in the youngest age classes (1–5 yo) is due to the high seroprevalence observed in these age groups (*Figure 1*) and also the fact that we do not allow for re-infections. As a consequence, only a small number of infections occur at older ages. Although the extremely high seropositivity rates at ages below 5 yo are not generally found with other enteroviruses, they have been reported for EV-D68 in other places (*Vogt and Crowe, 2018*), for example the US (*Harrison et al., 2019*), the Netherlands (*Karelehto et al., 2019*), and China (*Sun et al., 2018a*; *Xiang et al., 2017*). Clarifying the origin and the meaning of this high prevalence of neutralising antibodies at this young age (the role of serum neutralising antibodies in protection against infection and diseases) should be a priority for EV-D68 research (*Vogt and Crowe, 2018*). In particular, it is unclear whether cross-reactivity with other enteroviruses contributes to these high seropositivity and general increase in antibody titres with age (*Kamau et al., 2019*). If serum neutralising

antibodies are not a good correlate of protection against infection, the models may not capture well the age at which infections have increased, and it could be that the increase in infections is across age groups. Indeed, increased transmission may have also been associated with re-infections (and subsequent boosting of antibodies) in older age groups, consistent with the observed rise in geometric mean titres with age (*Kamau et al., 2019*). However, it seems unlikely that secondary infections can lead to paralysis, based on data for poliovirus.

Whether antigenic escape can explain re-infections in older children is not fully understood. There is evidence of amino acid changes in the BC and DE loop regions of the VP1 (which are thought to be epitopes for neutralising antibodies) that might have resulted in altered antigenic properties (*Dyrdak et al., 2019*; *Imamura et al., 2014*). However, although neutralisation assays conducted against different EV-D68 strains found some differences in neutralisation titres, study results have been inconsistent and their clinical and epidemiological significance unclear (*Kamau et al., 2019*; *Harrison et al., 2019*).

It is important to interpret well the results for the estimates of the FOI over time from our analysis under the assumptions of the models. First, as the best model uses a random walk on the FOI, the change in transmission that we infer happens continuously over several years. In reality, this may have occurred differently (e.g. in a shorter period of time). Our ability to recover more complex changes in transmission is limited by the data available. It would not be surprising if EV-D68 has exhibited biennial (or longer) cycles of transmission in England over the last few years, as it has been shown in the US (*Park et al., 2021*) and is common for other enteroviruses (*Pons-Salort and Grassly, 2018*). However, it is difficult to recover changes at this finer time scale with serology data unless sampling is very frequent (at least annual). Therefore, our study can only reveal broader long-term secular changes. Second, interpretation of the results before 2006 must be avoided for two reasons. On the one hand, as we go backwards in time, there is more uncertainty about the time of seroconversion of the individuals informing the estimates of the FOI. On the other hand, because age and time are confounded in cross-sectional seroprevalence measurements, the random walk on time may account for possible differences in the FOI through age (possibly higher in the youngest age classes, and lowest in the oldest), which are note explicitly accounted for here. This may explain the decline in FOI when going backwards in time before the first cross-sectional study in 2006. Third, allowing for a higher FOI in younger age classes would result in a shorter duration of maternal antibodies, which would make the results for the 1:16 cut-off more realistic in terms of decline of maternal antibodies. That said, these two parameters (rate of maternal antibodies decline and increase in FOI in younger age classes) would certainly be highly correlated and difficult to be jointly estimated.

Three major co-circulating EV-D68 clades (A–C) emerged globally in the 2000s (*Tokarz et al., 2012*) and have subsequently diversified, with only one monophyletic group (B1 and B3 genotypes) with a common ancestor in 2009 so far associated with AFM (*Hadfield et al., 2018*) (with the exception of one case associated with D1 [sometimes called A2] in 2018 in France; *Bal et al., 2019*). Individual viral lineages show rapid global spread, with recent outbreaks synchronised across Europe and the US representing circulation of the same dominant genotypes. Reported EV-D68 outbreaks in 2014 and 2016 were due to clade B viruses, while the 2018 outbreaks were reported to be linked to both B3 and A2 clade viruses in the UK (*The United Kingdom Acute Flaccid Paralysis (AFP) Task Force, 2019*), France (*Bal et al., 2019*), and elsewhere. In vitro studies of the neurotropism of these viruses compared with the ancestral strains have yielded conflicting results as to whether neurotropism has increased (*Hixon et al., 2019*; *Brown et al., 2018*; *Rosenfeld et al., 2019*). The timing of the increase in transmission estimated here (sometime before 2011) based on the analysis of the serology data may roughly correspond to the genetic emergence of clade B around 2007, and thus one could hypothesise that increased virus transmissibility is a trait associated with this clade. More efficient viral replication may enhance transmission as well as the probability of virus reaching the central nervous system, although changes in receptor usage could also play a role.

This work shows the value of modelling age-stratified seroprevalence data from consecutive cross-sectional studies in the understanding of the epidemiology of diseases caused by emerging human enteroviruses. The dynamics of most enterovirus serotypes over relatively long time scales have been shown to be driven by population immunity (*Pons-Salort and Grassly, 2018*). However, in rare instances, enterovirus serotypes have emerged as important causes of diseases after many years of circulation causing diseases at a much lower rate or even silently circulating. For example,

coxsackievirus A6 (CVA6) has emerged as the main serotype causing HFMD worldwide over the last decade (*Bian et al., 2015*). Finally, this work also shows the need to better understand and interpret individual serological data in terms of previous exposure and protection against infection and disease. This would help refining analytical approaches such as those used here to infer population-level processes.

# Materials and methods

## Serological data

We use data from three retrospective cross-sectional studies analysing serum samples representative of England's population in 2006 ($n$ = 516), 2011 ($n$ = 504), and 2017 ($n$ = 566) and available through the National Seroepidemiology Programme at Public Health England (*Osborne et al., 2000*). The neutralisation assay method and results from serological testing of the 2006 and 2017 sample sets have been previously described in *Kamau et al., 2019*. Neutralisation assays measured neutralising antibody titres against a B3 strain (*Kamau et al., 2019*; *Hadfield et al., 2018*), but Kamau et al. showed similar neutralisation effects across three different EV-D68 strains (*Kamau et al., 2019*).

## Statistical analysis

The FOI is the rate at which seronegative (susceptible) individuals become seropositive (infected). Cross-sectional age-stratified seroprevalence data can be used to estimate the FOI through so-called catalytic models (*Hens et al., 2010*). Catalytic models avoid modelling the dynamics of infected individuals directly; rather, they assume an unspecified mechanism that results in a seronegative individual becoming seropositive, its magnitude defining the FOI. For seroprevalence data, these models rely on the idea that the age and serostatus of an individual provide information on the probability of infection for the years between birth and the serosurvey.

We consider a serocatalytic model with maternal antibodies and no seroreversion (*Dighe, 2022*). Individuals enter the model at birth either seropositive due to the presence of maternal antibodies (m), or seronegative (n), with maternal antibodies declining at a constant rate $\omega$. Seronegative individuals become seropositive (p) at a rate $\lambda$, the FOI. We assume that after seroconversion, individuals cannot become seronegative again; this seems reasonable since seroprevalence does not show a decline through age (*Figure 1*). This model is described by the following system of differential equations:

$$\frac{dm}{da} = -\omega m$$

$$\frac{dn}{da} = \omega m - \lambda n$$

$$\frac{dp}{da} = \lambda n$$

which has the following analytical solution (*Dighe, 2022*):

$$m(a) = m_0 e^{-\omega a}$$

$$n(a) = m_0 \left( \frac{\omega}{\lambda - \omega} \left( e^{-\omega a} - e^{-\lambda a} \right) \right) + (1 - m_0) e^{-\lambda a}$$

$$p(a) = 1 - m(a) - n(a)$$

where $m_0$ is the proportion of individuals born with maternal antibodies, that we consider fixed and constant over time.

We are interested in estimating the rate of waning of maternal antibodies, $\omega$, and the FOI, $\lambda$. To link the catalytic model to the data from the serosurveys, we assume that the count of seropositive individuals within a serosurvey in year $t$ follows a binomial distribution:

$$z^{\tau}(t|c) \sim \text{binomial}\left(n^{\tau}(t), m^{\tau}(t) + p^{\tau}(t)\right),$$

where $n^{\tau}(t)$ indicates the sample size in year $t$ for individuals born in year $\tau$, $t \geq \tau$, as collected during the serosurvey, and $z^{\tau}(t|c)$ is the count of those who are seropositive for a given cut-off $c$. The

sum $m^\tau(t) + p^\tau(t)$ is the modelled proportion seropositive at year $t$ among those who were born in year $\tau$. Note that we further assume that testing uncovers seropositivity with 100% accuracy.

We test two different models representing different hypotheses about how the FOI changes over time $t$. Model 1 assumes a constant FOI over time,

$$\lambda(t) = \lambda,$$

And Model 2 allows the FOI to change over time following a random walk of order one:

$$\lambda_{t=t_1} \sim \text{normal}(0, 0.5)$$

$$\lambda_{t>t_1} \sim \text{normal}(\lambda_{t-1}, \sigma)$$

Both models assume a constant FOI through age.

The annual probability of infection, which is the proportion of the susceptible population that will become seropositive in a given year $t$, can be derived from the FOI: $p(t) = 1 - \exp(-\lambda(t))$.

Because seroprevalence in adults reaches almost 100% from about the 20-yo age class for a seropositivity cut-off of 1:16, and from the 30-yo for a more stringent cut-off of 1:64 (*Figure 1*), we fit the models to the data for age up to 40 yo.

The models were implemented in *Stan Development Team, 2020* and fitted to the data using MCMC. Four independent chains were simulated, each of 10,000 iterations, with a warmup of 3,000. Convergence was checked using the Rhat function.

We use the LOO metrics (*Vehtari et al., 2017*), implemented in the 'loo' R package (*Vehtari et al., 2023*), for model comparison. The LOO metrics aim to gauge how well a model generalises to an out-of-sample dataset and are an approximation to the explicit LOO cross-validation log-likelihood. In the explicit approach, the model is fitted to $n - 1$ datapoints and tested on a single hold-out datapoint and the log-likelihood on the test datapoint recorded. This approach is repeated for each of $n$ datapoints. The LOO is then effectively the average log-likelihood across all held-out datapoints. The 'loo' package uses Pareto-smoothed importance sampling to avoid explicitly refitting the model to the data $n$ times.

## Annual overall seroprevalence and reconstructed number of infections in each age class

Using the estimates of the FOI from the catalytic models above and data on population structure, we can estimate the overall (age-weighted) seroprevalence in the population each year $t$. We used data on the population structure in England for the years 1998, 2008, and 2018 from *Office for National Statistics, 2020*. The size of each age class for the years in between was obtained by linear interpolation.

Finally, we reconstructed the annual number of (new) infections in each age class using the estimates of the FOI and population structure data. To reconstruct the number of infections in a given year $t$ and age class $a$, we first reconstructed the proportion seronegative in the age class $a - 1$ until year $t - 1$, and then derived the proportion who would seroconvert during year $t$. We then multiplied that proportion by the population size of the corresponding age class and year.

Code and data are available through the GitHub repository: https://github.com/margapons/EV-D68_seroprevalence_England, (copy archived at *Pons-Salort, 2023*).

## Acknowledgements

We want to thank Sang Woo Park (Princeton University) for insightful comments to an initial version of the manuscript. MP-S is a Sir Henry Dale Fellow jointly funded by the Wellcome Trust and the Royal Society (grant number 216427/Z/19/Z). MP-S and NCG acknowledge funding from the MRC Centre for Global Infectious Disease Analysis (reference MR/R015600/1), jointly funded by the UK Medical Research Council (MRC) and the UK Foreign, Commonwealth & Development Office (FCDO), under the MRC/FCDO Concordat agreement and is also part of the EDCTP2 programme supported by the European Union. Work in PS's lab was supported by a Wellcome ISSF grant (ISSF204826/Z/16/Z). We thank the PHE Sero-Epidemiology Unit for access to residual samples for this public health investigation.

## Additional information

### Funding

| Funder | Grant reference number | Author |
|---|---|---|
| Wellcome Trust | 216427/Z/19/Z | Margarita Pons-Salort |
| Wellcome Trust | ISSF204826/Z/16/Z | Peter Simmonds |

The funders had no role in study design, data collection, and interpretation, or the decision to submit the work for publication. For the purpose of Open Access, the authors have applied a CC BY public copyright license to any Author Accepted Manuscript version arising from this submission.

### Author contributions

Margarita Pons-Salort, Conceptualization, Formal analysis, Investigation, Visualization, Methodology, Writing - original draft; Ben Lambert, Methodology, Writing - review and editing; Everlyn Kamau, Data curation, Writing - review and editing; Richard Pebody, Heli Harvala, Peter Simmonds, Resources, Writing - review and editing; Nicholas C Grassly, Writing - review and editing

### Author ORCIDs

Margarita Pons-Salort http://orcid.org/0000-0001-5597-9285

### Decision letter and Author response

Decision letter https://doi.org/10.7554/eLife.76609.sa1
Author response https://doi.org/10.7554/eLife.76609.sa2

## Additional files

### Supplementary files

• Supplementary file 1. Supplementary results. (**a**) Proportion of seropositive individuals in the 25–40 years old group. This proportion is used to fix the proportion of individuals born with maternal antibodies and is assumed constant across the study. (**b**) Model comparison for the two different datasets, corresponding to two different seropositivity cut-offs. For each model comparison, the first row corresponds to the model with the largest expected log pointwise density (ELPD), which measures the model expected predictive accuracy. For each model, its log-likelihood and the difference in the Bayesian leave-one-out (LOO) estimate of the ELPD compared to the best model are shown. (**c**) Model parameter estimates. Median and 95% credible intervals are presented.

• Transparent reporting form

### Data availability

Data analysed in this study are available through a GitHub repository.

The following dataset was generated:

| Author(s) | Year | Dataset title | Dataset URL | Database and Identifier |
|---|---|---|---|---|
| Pons-Salort M, Lambert B, Kamau E, Pebody R, Harvala H, Simmonds P, Grassly N | 2023 | EV-D68 seroprevalence in England | https://github.com/margapons/EV-D68_seroprevalence_England | GitHub, EV-D68_seroprevalence_England |

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
