## [Editor Report]

The authors use data from three cross-sectional age-stratified serosurveys on Enterovirus D68 from England between 2006 and 2017 to examine the transmission dynamics of this pathogen. This study's convincing methodology provides valuable insights into the changing dynamics of enterovirus D68, uncovering potential changes in the transmissibility of the virus. It will be of interest to infectious disease epidemiologists and surveillance professionals.

---

## [Decision Letter]

**Decision letter after peer review:**

Thank you for submitting your article "Changes in transmission of Enterovirus D68 (EV-D68) in England inferred from seroprevalence data" for consideration by *eLife*. Your article has been reviewed by 3 peer reviewers, and the evaluation has been overseen by a Reviewing Editor and a Senior Editor. The reviewers have opted to remain anonymous.

As is customary in *eLife*, the reviewers have discussed their critiques with one another. What follows below is the Reviewing Editor's edited compilation of the essential and ancillary points provided by reviewers in their critiques and in their interaction post-review. Please submit a revised version that addresses these concerns directly. Although we expect that you will address these comments in your response letter, we also need to see the corresponding revision clearly marked in the text of the manuscript. Some of the reviewers' comments may seem to be simple queries or challenges that do not prompt revisions to the text. Please keep in mind, however, that readers may have the same perspective as the reviewers. Therefore, it is essential that you attempt to amend or expand the text to clarify the narrative accordingly.

Essential revisions:

The authors use data from 3 cross-sectional age-stratified serosurveys on Enterovirus D68 from England between 2006 and 2017 to examine the transmission dynamics of this pathogen in this setting. Understanding these dynamics, including how it changes over time, may help uncover potential changes in the transmissibility of the virus. While the topic is relevant, interpretation of the results challenging largely due to the great uncertainty around how to interpret the serological (serostatus) data, and the impact this has on the inferences made. We ask the authors to perform some additional analyses and to provide more intuition to understand some of the key findings of this analysis.

1. We struggle to reconcile the evidence of a stable or even small drop in FoI after 2010 in the 1:64 models in contrast to the continued increase using the 1:16 cut-point.

2. It is hard to reconcile evidence of drops in FoI in the 1:64 (models 4 and 5 from 2010/11 (Figure 3)) with steadily increasing R0 in this period (Figure 4). Is this due to changes in the susceptibility proportion. It would be good to understand if there are important assumptions in the Farrington approach that may also contribute to this discrepancy.

3. One of the major findings of the paper is that there is a steadily increasing R0 (using the 1:64 cut-point). This again is difficult to understand and would suggest there are either year on year increases in inherent transmissibility of the virus through fitness changes, or year on year increases in the mixing of the population. It would be useful for the authors to discuss potential explanations for an inferred gradual increase in R0.

4.The estimated FOI in 1 year olds is very very high (with a suggestion that up to 75% get infected within a year) and difficult to believe, especially as the force of infection is assumed much lower for all other ages. The authors exclude all <1s due to maternal antibodies, which seems sensible, however, does this mean that it is impossible for <1s to become infected in the model? We know for other pathogens (e.g., dengue virus) with protection from maternal antibodies that the protection from infection is gone after a few months. Maybe allowing for infections in the first year of life too would reduce the very large, and difficult to believe, difference in risk between 1 year olds and older age groups. I suspect you wouldn't need to rely on <1 serodata – just allow for infections in this time period.

5. Relatedly would it be possible to break the age data into months rather than years in these infants to help tease apart what happens in the critical early stages of life.

6. Additional context of EV-D68 in the study setting of England would be useful. While the Introduction does mention AFM cases "in the UK and elsewhere in Europe" (line 53), a summary of reported data on EV-D68/AFM in England prior to this study would provide important context. The Methods refers to "whether transmission had increased over time (before the first reported big outbreak of EV-D68 in the US in 2014)" (lines 133-134), rather than in this setting. It would be useful to summarize the viral genomic data from the region for additional context – particularly since the emergence of a viral clade is highlighted as a co-occurrence with the increased transmissibility detected in this analysis.

7. Given the substantial uncertainty in the assay, it seems optimistic to attempt to fit annual force of infections in the 30 year period prior to the start of the sampling periods. Authors should consider including a constant λ prior to the dates of the first study across the models considered.

8. While the authors have made data sets available, it would be good to make computer code available as well.

*Reviewer #1 (Recommendations for the authors):*

In the abstract it would be helpful to have some info on the AFM in England as a link between the global picture explained and then this analysis which is for England.

Line 188-120: I agree with the point here, but wonder if a little more to be added to help guide the reader through this thinking from lower seroprevelance to age. I also wonder if it isn't due to an increase in transmission what is it due to? Perhaps this could also be elaborated on.

Line 169 (and methods): Please provide more information on the LOO criteria and what was left out. More required in the main text and also in the methods.

*Reviewer #2 (Recommendations for the authors):*

– The submission lists only 2 contributing authors, but the manuscript lists additional authors. The author lists should be synced.

– While the authors have made data sets available, computer code was not available as far as I could tell.

– For Models 4 and 5: what were the estimated values of sigma0 and σ? They were not included in Table S1. In the Methods section, λ_{t=t1} is modeled as a Normal centered at 0 – is this on the log scale?

– Figure 2 (E and F): what does the purple class indicate for this model? Is it an average across all other age classes?

– Table 1: it was not clear why Model 3's δ LOO is so poor compared to Model 5 despite the similar visual fits of the models (Figure 3 vs. Figure S5), particularly among the under 20 year-olds. Could the authors provide some more intuition on this? Are there particular data points that are highly influential on the LOO statistic?

– Line 142: should this sentence also include γ?

– Lines 159-167: the values cited in the text do not seem to match those in Table S1.

– Line 172: what do these p-values represent?

[Editors' note: further revisions were suggested prior to acceptance, as described below.]

Thank you for resubmitting your work entitled "Changes in transmission of Enterovirus D68 (EV-D68) in England inferred from seroprevalence data" for further consideration by *eLife*. Your revised article has been evaluated by Eduardo Franco (Senior Editor) and a Reviewing Editor.

The manuscript has been improved, but there are a couple of issues that need to be addressed, as outlined below (see comments from reviewer #4)

*Reviewer #4 (Recommendations for the authors):*

I have two comments on the revision:

1. I agree with the authors' decision to implement maternal antibodies as part of their modeling approach. However, the estimated proportion of individuals with maternal antibodies by age seems very high for the 1:16 cutoff. Is it realistic to have maternal antibodies in >25% of 2 year olds? If not, it might be prudent to have m(a) go to zero by a certain age.

2. I had made a comment in the previous round of review about extending the x-axis to the start of the time period of estimation: this was in reference to FOI, not seroprevalence. The FOI estimates in Figure 3A begin in 1990, but the oldest cohort in this analysis are 40 y in 2006, and it's not clear what is assumed about FOI between 1966 and 1990. Or does the random walk on the FOI begin in 1966? It would be good to show those results.

---

## [Author Response]

Essential revisions:The authors use data from 3 cross-sectional age-stratified serosurveys on Enterovirus D68 from England between 2006 and 2017 to examine the transmission dynamics of this pathogen in this setting. Understanding these dynamics, including how it changes over time, may help uncover potential changes in the transmissibility of the virus. While the topic is relevant, interpretation of the results challenging largely due to the great uncertainty around how to interpret the serological (serostatus) data, and the impact this has on the inferences made. We ask the authors to perform some additional analyses and to provide more intuition to understand some of the key findings of this analysis.

We have changed the order of the comments, to describe first the main changes to the manuscript and make the explanations clearer.

5. Relatedly would it be possible to break the age data into months rather than years in these infants to help tease apart what happens in the critical early stages of life.

Yes. We have added two figures (new Figures 1C and 1D) showing the prevalence of antibodies in children <1 yo. We show these data for the three serosurveys combined, because the number of individuals per month of age is very small.

4.The estimated FOI in 1 year olds is very very high (with a suggestion that up to 75% get infected within a year) and difficult to believe, especially as the force of infection is assumed much lower for all other ages. The authors exclude all <1s due to maternal antibodies, which seems sensible, however, does this mean that it is impossible for <1s to become infected in the model? We know for other pathogens (e.g., dengue virus) with protection from maternal antibodies that the protection from infection is gone after a few months. Maybe allowing for infections in the first year of life too would reduce the very large, and difficult to believe, difference in risk between 1 year olds and older age groups. I suspect you wouldn't need to rely on <1 serodata – just allow for infections in this time period.

We thank the reviewers for this important suggestion. We have changed the catalytic models and now use a model that includes individuals with maternal antibodies at birth. With this new model, we no longer need to make extra hypothesis about differences in FOI at age 1. We now estimate both the rate of decline of maternal antibodies and the FOI (that we assume constant through age). We have substantially revised the Methods section to describe the new approach, updated all the results accordingly. These includes the model fit (new Figure 2) and the estimates of the FOI over time (new Figure 3A). We also show the decline of the estimated proportion of individuals with maternal antibodies through age (new Figure 3B).

1. We struggle to reconcile the evidence of a stable or even small drop in FoI after 2010 in the 1:64 models in contrast to the continued increase using the 1:16 cut-point.

With the new model (i.e. with maternal antibodies) the difference in the “trajectory” of the FOI over time for the two cut-off persists. We think this is driven by the difference in seroprevalence between the youngest age classes in 2011 vs. 2017 for the two cut-offs (see Figures 1A and 1B). For the 1:16 cut-off, seroprevalence in the young age classes increases over time between the first serosurvey in 2006 and the last one in 2017 (Figure 1A). However, for the 1:64 cut-off, seroprevalence in the young age classes is very similar for the 2011 and 2017 serosurveys. The serocatalytic models therefore do not estimate a change in the FOI between those two time points.

We thank the reviewers for highlighting this point, which we did not discuss before. We have now added the following text to the Results:

“The best model (Model 2) estimated an increase in transmission over time during the study period (2006-2017) for both seropositivity cut-offs, as shown by the estimated FOI in Figure 3A. For the cut-off of 1:16, the FOI continued to increase until the end of the study period, in 2017. However, for the more stringent cut-off of 1:64, the FOI plateaued from around 2011. These differences reflect the differences in seroprevalence observed in the young age classes (1 – 20 yo) between 2011 and 2017 for the two cut-offs (Figure 1).”

And we have also added the following paragraph to the Discussion presenting the main differences in the results for the two cut-offs:

“Although our results of an increase in FOI over the study period are robust to the choice of the seropositivity cut-off, we find striking differences in terms of the magnitude of the FOI and the decline of maternal antibodies depending on the seropositivity cut-off used. The results obtained with the more stringent cut-off of 1:64 seem more realistic, both for the estimated annual probabilities of infection (range 0.32-0.52 for the 1:16, and 0.13-0.20 for the 1:64, for the period 2006-2017) and rate of decline of maternal antibodies (average duration of maternal antibodies around 18 mo for the 1:16, and 5 mo for the 1:64). Furthermore, the analysis with the 1:64 cut-off infers that there has not been significant changes in the FOI since 2011; however, with the 1:16 cut-off, the FOI continuous to increase until 2017. These results therefore raise the question of what is a suitable seropositivity cut-off to define previous exposure to EV-D68. That said, most EV-D68 seroepidemiology studies published to date present results for a cut-off of 1:8 (as classically used for polioviruses) or 1:16 (17, 19-21).”

2. It is hard to reconcile evidence of drops in FoI in the 1:64 (models 4 and 5 from 2010/11 (Figure 3)) with steadily increasing R0 in this period (Figure 4). Is this due to changes in the susceptibility proportion. It would be good to understand if there are important assumptions in the Farrington approach that may also contribute to this discrepancy.

We have removed the estimates of R0 from the manuscript and only present the reconstruction of the annual number of new infections per age class and year (new Figure 5). We think this measure is more adapted to the discussion of the results.

In addition, when using the classical expression R{0t}=1/(1-S(t)), with S(t) the annual proportion seropositive, the high seroprevalence estimates (new Figure 4) result in extremely high estimates of the basic reproduction number (median ranges: 11.6 – 29.7 for 1:16 and 3.3 – 7.6 for 1:64 during the period 2006 to 2017).

We had previously used the Farrington approach as it is adapted to cases when the force of infections is different for different age classes.

3. One of the major findings of the paper is that there is a steadily increasing R0 (using the 1:64 cut-point). This again is difficult to understand and would suggest there are either year on year increases in inherent transmissibility of the virus through fitness changes, or year on year increases in the mixing of the population. It would be useful for the authors to discuss potential explanations for an inferred gradual increase in R0.

We have removed the estimates of R0 from the manuscript.

6. Additional context of EV-D68 in the study setting of England would be useful. While the Introduction does mention AFM cases "in the UK and elsewhere in Europe" (line 53), a summary of reported data on EV-D68/AFM in England prior to this study would provide important context. The Methods refers to "whether transmission had increased over time (before the first reported big outbreak of EV-D68 in the US in 2014)" (lines 133-134), rather than in this setting. It would be useful to summarize the viral genomic data from the region for additional context – particularly since the emergence of a viral clade is highlighted as a co-occurrence with the increased transmissibility detected in this analysis.

We have added a figure (new Figure 1 —figure supplement 1) showing the annual number of EV-D68 detections reported by Public Health England from 2004 to 2020.

We have also added the following text to the introduction: “Similarly, in the UK, reported EV-D68 virus detections also show a biennial pattern between 2014 and 2018 (Figure 1 —figure supplement 1).”

We have also amended the sentence in the Methods.

Finally, Author response image 1 is a screenshot of the nexstrain tree for EV-D68 based on the VP1 region and with tips representing sequences from the UK (light blue) and European countries in colour. There is a lot of mixing between sequences from different regions, indicating widespread transmission and small regional clustering. We have added the following text to the Discussion: “Reported EV-D68 outbreaks in 2014 and 2016 were due to clade B viruses, while the 2018 outbreaks were reported to be linked to both B3 and A2 clade viruses in the UK (10), France (32) and elsewhere.”

**Author response image 1. sa2fig1:** 

7. Given the substantial uncertainty in the assay, it seems optimistic to attempt to fit annual force of infections in the 30 year period prior to the start of the sampling periods. Authors should consider including a constant λ prior to the dates of the first study across the models considered.

We thank the reviewers for the suggestion.

We implemented this change (constant FOI before 2006) in the previous models without maternal antibodies and the result for the random-walk-based models was that the variance of the random walk was estimated over a very short period, thus resulting in a rather nonsmoothed FOI.

Implementing this change with the new models with maternal antibodies and random-walk on the FOI was technically a bit complex. We therefore kept the simple random-walk over the whole period and added the following paragraph to the Discussion:

“It is important to interpret well the results for the estimates of the FOI over time from our analysis under the assumptions of the models. First, as the best model uses a random walk on the FOI, the change in transmission that we infer happens continuously over several years. In reality, this may have occurred differently (e.g. in a shorter period of time). Our ability to recover more complex changes in transmission is limited by the data available. It would not be surprising if EV-D68 has exhibited biennial (or longer) cycles of transmission in England over the last few years, as it has been shown in the US (7) and is common for other enteroviruses (30). However, it is difficult to recover changes at this finer time scale with serology data unless sampling is very frequent (at least annual). Therefore, our study can only reveal broader long-term secular changes. Second, interpretation of the results before 2006 must be avoided for two reasons. On the one hand, as we go backwards in time, there is more uncertainty about the time of seroconversion of the individuals informing the estimates of the FOI. On the other hand, because age and time are confounded in cross sectional seroprevalence measurements, the random walk on time may account for possible differences in the FOI through age (possibly higher in the youngest age classes, and lowest in the oldest), which are note explicitly accounted for here. This may explain the decline in FOI when going backwards in time before the first cross-sectional study in 2006.”

8. While the authors have made data sets available, it would be good to make computer code available as well.

The code is available in the GitHub repo: https://github.com/margapons/EVD68_seroprevalence_England

Reviewer #1 (Recommendations for the authors):In the abstract it would be helpful to have some info on the AFM in England as a link between the global picture explained and then this analysis which is for England.

Data for AFM is scarce for England and publications do not always allow to discern the overlap of cases reported nor to differentiate between cases of AFP (acute flaccid paralysis) and AFM (acute flaccid myelitis) – the last one being defined only in 2014. A ballpark estimate for the number of AFM cases in the UK is in the range of 20-30, between January 2015 and December 2018, based on data reported in publications (1-5) below. As this number is not precise, nor confirmed by health authorities, we prefer not to report it. Note, however, that now we do report number of EV-D68 detections in the UK.

1. Kirolos, A., et al., Outcome of paediatric acute flaccid myelitis associated with enterovirus D68: a case series. Dev Med Child Neurol, 2019. 61(3): p. 376-380.

2. Cottrell, S., et al., Prospective enterovirus D68 (EV-D68) surveillance from September 2015 to November 2018 indicates a current wave of activity in Wales. Euro Surveill, 2018. 23(46).

3. Williams, C.J., et al., Cluster of atypical adult Guillain-Barre syndrome temporally associated with neurological illness due to EV-D68 in children, South Wales, United Kingdom, October 2015 to January 2016. Euro Surveill, 2016. 21(4).

4. Bubba, L., et al., Circulation of non-polio enteroviruses in 24 EU and EEA countries between 2015 and 2017: a retrospective surveillance study. Lancet Infect Dis, 2020. 20(3): p. 350-361.

5. The United Kingdom Acute Flaccid Paralysis Afp Task, F., An increase in reports of acute flaccid paralysis (AFP) in the United Kingdom, 1 January 2018-21 January 2019: early findings. Euro Surveill, 2019. 24(6).

Line 188-120: I agree with the point here, but wonder if a little more to be added to help guide the reader through this thinking from lower seroprevelance to age. I also wonder if it isn't due to an increase in transmission what is it due to? Perhaps this could also be elaborated on.

Thanks for this suggestion. We agree with the reviewer that the previous formulation was unclear. These now reads:

“Age-stratified seroprevalence was generally lower in 2006 compared to 2011 and 2017, which suggests individuals acquired their first infection at a lower age through the study period, leading to a decrease in the mean age of exposure. This could potentially be consistent with increased transmission (e.g. through increased virus fitness or the accumulation of susceptible) or other mechanisms (such as a change in the virus to have a higher tendency to infect children).”

Line 169 (and methods): Please provide more information on the LOO criteria and what was left out. More required in the main text and also in the methods.

The LOO metrics aim to gauge how well a model generalises to an out-of-sample dataset and are an approximation to the explicit leave-one out cross-validation log-likelihood (Vehtari, Gelman and Gabry, 2017). In the explicit approach, the model is fitted to n-1 datapoints and tested on a single hold-out datapoint and the log-likelihood on the test datapoint recorded. This approach is repeated for each of n datapoints. The LOO is then effectively the average log-likelihood across all held-out datapoints (accounting for uncertainty in the posterior).

We used the “loo” R package (Vehtari et al., 2022) to calculate the LOO metric, which uses Pareto-smoothed importance sampling to avoid explicitly refitting to the data n times. This metric and the method for its approximation are widely used in applied Bayesian inference for performing model comparison.

Vehtari A, Gelman A, and Gabry J. Practical Bayesian model evaluation using leave-one-out cross-validation and WAIC. *Statistics and computing* (2017): 1413-1432.

Vehtari A, Gabry J, Magnusson M, Yao Y, Bürkner P, Paananen T, Gelman A (2022). “loo: Efficient leave-one-out cross-validation and WAIC for Bayesian models.” R package version 2.5.1, https://mc-stan.org/loo/

We have now added the following text to the Methods:

We use the LOO metrics (39), implemented in the “loo” R package (40), for model comparison. The LOO metrics aim to gauge how well a model generalises to an out-of-sample dataset and are an approximation to the explicit leave-one-out cross-validation loglikelihood. In the explicit approach, the model is fitted to n-1 datapoints and tested on a single hold-out datapoint and the log-likelihood on the test datapoint recorded. This approach is repeated for each of n datapoints. The LOO is then effectively the average loglikelihood across all held-out datapoints. The “loo” package uses Pareto-smoothed importance sampling to avoid explicitly refitting the model to the data n times.

Reviewer #2 (Recommendations for the authors):– The submission lists only 2 contributing authors, but the manuscript lists additional authors. The author lists should be synced.

The manuscript has 7 authors (listed in the pdf file). It is unclear to us why the submission showed only two.

– While the authors have made data sets available, computer code was not available as far as I could tell.

The code is available in the GitHub repo https://github.com/margapons/EVD68_seroprevalence_England

– For Models 4 and 5: what were the estimated values of sigma0 and σ? They were not included in Table S1. In the Methods section, λ_{t=t1} is modeled as a Normal centered at 0 – is this on the log scale?

We have now conducted a sensitivity analysis and seen that the value of sigma0 does not affect the estimates. We therefore fix sigma0 = 0.5, and only estimate σ. The Methods section has been updated accordingly and we also provide the estimates of σ and other parameters in the new Table S3.

The prior of λ_{t=t1} is a Normal distribution centered at 0 (in the natural scale; not log scale) restricted to positive values (a “half-normal” distribution). It is effectively truncated at zero, as we set the lower bound for this parameter at zero.

– Figure 2 (E and F): what does the purple class indicate for this model? Is it an average across all other age classes?

This comment does no longer apply, as the models have changed.

– Table 1: it was not clear why Model 3's δ LOO is so poor compared to Model 5 despite the similar visual fits of the models (Figure 3 vs. Figure S5), particularly among the under 20 year-olds. Could the authors provide some more intuition on this? Are there particular data points that are highly influential on the LOO statistic?

This comment does no longer apply, as the models have changed.

– Line 142: should this sentence also include γ?

This comment does no longer apply, as the models have changed and there is no parameter γ.

– Lines 159-167: the values cited in the text do not seem to match those in Table S1.

This comment does no longer apply, as the models have changed.

– Line 172: what do these p-values represent?

These p-values represent the probability that the null hypothesis that two models provide the same fit to the data is true (vs the alternative that one model outperforms the other). The p-values are calculated under a normal approximation of the ELPD measure of fit as discussed in (Vehtari, Gelman and Gabry, 2017).

As the differences in ELPD between the two new models (constant and the random-walk) are large for the two datasets, we no longer present these p-values, which we understand may not be straightforward to interpret for the reader.

[Editors' note: further revisions were suggested prior to acceptance, as described below.]

The manuscript has been improved, but there are a couple of issues that need to be addressed, as outlined below (see comments from reviewer #4)Reviewer #4 (Recommendations for the authors):I have two comments on the revision:1. I agree with the authors' decision to implement maternal antibodies as part of their modeling approach. However, the estimated proportion of individuals with maternal antibodies by age seems very high for the 1:16 cutoff. Is it realistic to have maternal antibodies in >25% of 2 year olds? If not, it might be prudent to have m(a) go to zero by a certain age.

We agree with the reviewer that the estimate of >25% of individuals with maternal antibodies by the age of 2 yo obtained for the 1:16 cut-off seems unrealistic. Actually, we believe that all the results (including also FOI estimates) are more realistic for the 1:64 cutoff than for the 1:16, and we already discuss this in the second paragraph of the Discussion. We think it is important to show the results obtained with both cut-offs, as studies reporting EVD68 seroprevalence published so far have used a seropositivity cut-off of 1:8 or 1:16, which we think may not be the most suitable for this enterovirus serotype.

A main observation of our and other studies is that seroprevalence is very high in the 12-23 months old for the 1:16 cut-off and this is discussed at several points in the Discussion. In our approach, the rate of decline of maternal antibodies (parameter omega) is estimated, and therefore, informed by the seroprevalence data. If we fixed it and “forced it to go to zero by a certain age”, as suggested by the reviewer, we would need to consider a higher FOI in the younger age classes to be able to explain the high seroprevalence observed in those. We have now added the following sentence to the Discussion to acknowledge the limitation:

“Third, allowing for a higher FOI in younger age classes would result in a shorter duration of maternal antibodies, which would make the results for the 1:16 cut-off more realistic in terms of decline of maternal antibodies. That said, these two parameters (rate of maternal antibodies decline and increase in FOI in younger age classes) would certainly be highly correlated and difficult to be jointly estimated.”

2. I had made a comment in the previous round of review about extending the x-axis to the start of the time period of estimation: this was in reference to FOI, not seroprevalence. The FOI estimates in Figure 3A begin in 1990, but the oldest cohort in this analysis are 40 y in 2006, and it's not clear what is assumed about FOI between 1966 and 1990. Or does the random walk on the FOI begin in 1966? It would be good to show those results.

The random walk starts at 1966. The figure with the complete estimate of the FOI over time is shown in Author response image 2:

We prefer not to show the FOI estimates for the initial period for two reasons. First, we are interested on seeing whether there has been an increase in transmission in the few years before the first large EV-D68 outbreaks in 2014. This is, between a few years before 2006 (our first time point with data) and 2014. Second, because our first time point of observations is 2006 and seroprevalence increases quickly with age, the FOI estimates going backwards in time are informed by older, mostly seropositive individuals, which results in a basically flat or not well estimated FOI. This is clearer for the 1:16 cut-off, for which we observe a large increase in the credible intervals. This is because seroprevalence quickly reaches 100% (around the age of 20) and therefore, there is not much signal in the data to inform the FOI going backwards in time, i.e. before 2006-20=1986 approximately.